# Energy Stability Property of the CPR Method Based on Subcell Second-Order CNNW Limiting in Solving Conservation Laws

**DOI:** 10.3390/e25050729

**Published:** 2023-04-28

**Authors:** Ran Liu, Zhen-Guo Yan, Huajun Zhu, Feiran Jia, Xinlong Feng

**Affiliations:** 1School of Mathematics and Systems Science, Xinjiang University, Urumqi 830017, China; 2State Key Laboratory of Aerodynamics, China Aerodynamics Research and Development Center, Mianyang 621000, China; 3School of Power and Energy, Northwestern Polytechnical University, Xi’an 710000, China

**Keywords:** conservation laws, correction procedure via reconstruction (CPR), second-order compact nonuniform nonlinear weighted (CNNW2) scheme, energy stability

## Abstract

This paper studies the energy stability property of the correction procedure via reconstruction (CPR) method with staggered flux points based on second-order subcell limiting. The CPR method with staggered flux points uses the Gauss point as the solution point, dividing flux points based on Gauss weights, with the flux points being one more point than the solution points. For subcell limiting, a shock indicator is used to detect troubled cells where discontinuities may exist. Troubled cells are calculated by the second-order subcell compact nonuniform nonlinear weighted (CNNW2) scheme, which has the same solution points as the CPR method. The smooth cells are calculated by the CPR method. The linear energy stability of the linear CNNW2 scheme is proven theoretically. Through various numerical experiments, we demonstrate that the CNNW2 scheme and CPR method based on subcell linear CNNW2 limiting are energy-stable and that the CPR method based on subcell nonlinear CNNW2 limiting is nonlinearly stable.

## 1. Introduction

Computational fluid dynamics (CFD) is widely used in industry, and corresponding CFD commercial softwares have been developed as well. However, most of these software programs use low-order schemes, which have more dissipation errors. Compared with low-order schemes, high-order schemes offer better accuracy and lower computational costs achieving the same error level, especially for problems with complex physics and geometry [1].

In 2007, Huynh [2] introduced a high-order method for solving hyperbolic conservation laws called the flux reconstruction (FR) method. The basic idea of this method is to collocate the solution points and the flux points on the cell, construct the solution polynomial and the flux polynomial by Lagrange interpolations, then use a numerical flux at the cell interface to correct the divergence of the flux polynomial. In the linear case, by selecting the $g_{DG}$ correction function, the FR method can recover the discontinuous Galerkin (DG) [3,4,5,6] method. From the FR method, the spectral difference (SD) [7,8,9,10] method can be recovered by selecting the $g_{SD}$ correction function. In 2009, Wang and Gao [11,12] extended the FR method to two-dimensional (2D) triangle grids and hybrid grids and proposed the lifting collocation penalty (LCP) method. The FR and LCP methods are closely related, and are collectively referred to as the correction procedure via reconstruction (CPR) method.

In 2011, Vincent et al. [13] constructed an energy-stable FR method for a one-dimensional (1D) scalar convection equation. This method is controlled by a scalar parameter *c*; when *c* is within a certain range, the method is energy-stable, which is called the energy stable flux reconstruction (ESFR) method. The energy stability of the 1D linear convection equation has been proven. Subsequently, Jameson et al. [14] studied the stability of the ESFR method for a 1D nonlinear equation, and their results showed that aliasing errors are introduced in the discretization of nonlinear fluxes, potentially leading to instability. In 2015, Sheshadri et al. [15] studied the energy stability of the 2D linear convection equation on Cartesian grids, with the results showing that for uniform grids the energy is stable when the parameter *c* is non-negative. Spiegel et al. [16] studied the de-aliasing strategy of the FR method, which is called the over-integration method. This strategy significantly improves the stability of the FR method.

Because the CPR method is a high-order linear scheme, it tends to produce spurious numerical oscillations when solving strong discontinuities. Therefore, it is necessary to develop a shock capturing strategy suitable for the CPR method. In 2014, Sonntag and Munz [17] proposed a shock capturing algorithm based on the high-order DG method; in this method, the shock regions are divided into several subcells and calculated by a finite volume scheme. This algorithm combines the favourable characteristics of the DG method in smooth regions and the TVD finite volume method in discontinuous regions. Dumbser et al. [18,19] proposed a simple and robust posterior subcell finite volume limiter for the DG method on unstructured grids. The idea of this posterior method is to generate the candidate solution first, then go back to previous time step for correction if elements do not satisfy the posteriori detection criteria. In 2021, Kochi and Ramakrishna [20] proposed a compact subcell weighted essentially non-oscillatory (CSWENO) limiting strategy for the DG method. In 2022, Zhu et al. [21] proposed a CPR method with a subcell shock capturing strategy based on the Gauss solution points and staggered flux points. The troubled cells are detected by a shock indicator and solved by compact nonuniform nonlinear weighted (CNNW) schemes, and the smooth cells are solved by the CPR method with staggered flux points. Shi et al. [22] extended the CPR method with subcell second-order CNNW (CNNW2) limiting to unstructured quadrilateral grids. Liu et al. studied the CPR method with staggered flux points, and found that this method exhibits nonlinear stability with the $g_{DG}$ correction function [23].

In this paper, we investigate the energy stability of the CPR method with staggered flux points based on subcell CNNW2 limiting. The main contributions of this work are as follows:

1. The energy stability of the linear CNNW2 scheme is proved theoretically. By using $L^{2}$ energy estimation method [24], the change rate of the energy norm of the numerical solutions constructed by the CNNW2 scheme does not increase over time.

2. Various numerical experiments based on the linear advection equation and Euler equations are conducted. The results show that the CPR method with staggered flux points, the linear scheme of CNNW2, and the CPR method with staggered flux points based on subcell linear CNNW2 limiting are linearly energy stable. In addition, the CPR method with staggered flux points based on subcell nonlinear CNNW2 limiting is nonlinearly stable.

This paper is organized as follows. In Section 2, the governing equations and the CPR method with staggered flux points for 1D conservation laws are briefly introduced. In Section 3, the CPR method with staggered flux points based on subcell CNNW2 limiting is provided, and a linear energy stability analysis of the linear CNNW2 method is presented. In Section 4, the numerical results for a series of test cases are presented in detail. The concluding remarks are provided in Section 5. Finally, we prove the energy stability property of the first-order CNNW2 scheme in Appendix A.

## 2. Governing Equations and Discretization Methods

### 2.1. Governing Equations

Consider the 2D Euler equations [22] in the conservation form,
(1)$$\frac{\partial U}{\partial t}+\frac{\partial F}{\partial x}+\frac{\partial G}{\partial y}=0,$$
where U is the vector of the conservative variables, F and G are the inviscid fluxes.
(2)$$U=\left[\begin{array}{c} \rho \\ \rho u\\ \rho v\\ E \end{array}\right],F=\left[\begin{array}{c} \rho u\\ \rho u^{2}+p\\ \rho uv\\ u(E+p) \end{array}\right],G=\left[\begin{array}{c} \rho v\\ \rho uv\\ \rho v^{2}+p\\ v(E+p) \end{array}\right],$$
(3)$$E=\frac{p}{\gamma -1}+\frac{1}{2}\rho \left(u^{2}+v^{2}\right),$$
where ρ is the density, *u* and *v* are the velocity components, *p* is the pressure, and *E* is the total energy. For an ideal gas, the specific heat ratio is γ = 1.4.

### 2.2. CPR Method with Staggered Flux Points

Consider the 1D scalar convection equation
(4)$$\frac{\partial u}{\partial t}+\frac{\partial f}{\partial x}=0,$$
where *u* is the variable and *f* is the flux. Firstly, the equation is transformed from a physical cell $E_{n}$ to a computational cell with the interval *I* = [−1, 1]. As shown in Figure 1, the red circles represent the solution points. In this paper, we consider a linear transformation, where the linear relationship that maps *I* onto $E_{n}$ and its inverse are [2]
(5)$$x\left(\xi \right)=\frac{x_{n}+x_{n+1}}{2}+\xi h/2\;\;\mathrm{and}\;\;\xi \left(x\right)=2\left(x-\frac{x_{n}+x_{n+1}}{2}\right)/h$$
where, h is the length of the physical cell.

Upon transformation to computational cells. The CPR method for Equation (Equation 4) becomes
(6)$$\frac{\partial {\hat{u}}^{\delta }}{\partial t}+\frac{\partial {\hat{f}}^{\delta }}{\partial \xi }=0,$$
where ${\hat{u}}^{\delta }=u^{\delta }$, ${\hat{f}}^{\delta }=\frac{hf^{\delta }}{2}$, $\frac{h}{2}$ is the Jacobian, and *h* is the length of the cell.

For each cell, the (k−1)-order approximate solution polynomial is constructed by a Lagrange interpolation as
(7)$${\hat{u}}^{\delta }=\sum_{i=1}^{k}{\hat{u_{i}}}^{\delta }l_{i},$$
where ${\hat{u_{i}}}^{\delta }$ is the state variable at the solution point $\xi_{i}$ and $l_{i}$ is the (k−1)-order Lagrange basis function, which has the following form:(8)$$l_{i}=\prod_{j=1,j\neq i}^{k}\left(\frac{\xi -\xi_{j}}{\xi_{i}-\xi_{j}}\right).$$

In addition, ${\hat{f}}^{\delta D}$ is constructed in a similar way:(9)$${\hat{f}}^{\delta D}=\sum_{i=1}^{k}{\hat{f_{i}}}^{\delta D}l_{i},$$
where ${\hat{f_{i}}}^{\delta D}$ is the flux at the solution point $\xi_{i}$.

Finally, by introducing the numerical flux and correction function, the spatial semi-discrete scheme of the original CPR [2] is obtained as
(10)$$\frac{\partial {\hat{u_{i}}}^{\delta }}{\partial t}=-\left(\sum_{i=1}^{k}{\hat{f_{i}}}^{\delta D}\frac{dl_{i}}{d\xi }\left(\xi_{i}\right)+\left({\hat{f_{L}}}^{\delta I}-{\hat{f_{L}}}^{\delta D}\right)\frac{dg_{L}}{d\xi }\left(\xi_{i}\right)+\left({\hat{f_{R}}}^{\delta I}-{\hat{f_{R}}}^{\delta D}\right)\frac{dg_{R}}{d\xi }\left(\xi_{i}\right)\right),$$
where *L* and *R* represent the left and right interfaces of the cell, respectively, $g_{L}$ and $g_{R}$ are correction functions of order *k*, and ${\hat{f}}^{\delta I}$ is the numerical flux. The solution points and flux points for the original CPR method are shown in Figure 2. The solution points of this method coincides with the flux points.

To simplify the description, the CPR method with staggered flux points and the original CPR method are named the CPR (Q>P) and CPR (Q=P) methods, respectively. Here, *Q* represents the number of flux points, and *P* represents the number of solution points. The difference between the CPR (Q>P) and CPR (Q=P) methods is the selection of flux points, resulting in different ${\hat{f}}^{\delta D}$. The solution points and the flux points for the CPR (Q>P) method are shown in Figure 3. It should be noted that the solution points of the CPR (Q>P) method are interleaved with the flux points, and that the lengths of the subcells are determined by the Gauss quadrature weights. The difference is as follows.

The *k*th−order flux polynomial is constructed by the Lagrange interpolation, and the form is as follows:(11)$${\hat{f}}^{\delta D}=\sum_{i=1}^{k+1}{\hat{f_{i}}}^{\delta D}l_{i}^{f},$$
where ${\hat{f_{i}}}^{\delta D}$ is the flux at the flux point $\xi_{i}^{f}$ and where $l_{i}^{f}$ is the *k*th−order Lagrange basis function, which has the following form:(12)$$l_{i}^{f}=\prod_{j=1,j\neq i}^{k+1}\left(\frac{\xi -\xi_{j}^{f}}{\xi_{i}^{f}-\xi_{j}^{f}}\right).$$

Finally, the spatial semi-discrete scheme of CPR (Q>P) is obtained as
(13)$$\frac{\partial {\hat{u_{i}}}^{\delta }}{\partial t}=-\left(\sum_{i=1}^{k+1}{\hat{f_{i}}}^{\delta D}\frac{dl_{i}^{f}}{d\xi }\left(\xi_{i}\right)+\left({\hat{f_{L}}}^{\delta I}-{\hat{f_{L}}}^{\delta D}\right)\frac{dg_{L}}{d\xi }\left(\xi_{i}\right)+\left({\hat{f_{R}}}^{\delta I}-{\hat{f_{R}}}^{\delta D}\right)\frac{dg_{R}}{d\xi }\left(\xi_{i}\right)\right).$$

Note that the CPR (Q=P) method using the $g_{DG}$ correction function belongs to the class of ESFR methods, which was originally constructed by Vincent for 1D linear convection equations [13]. Similarly, the CPR (Q>P) method under this correction function is a version of the ESFR (Q>P) method. The nonlinear stability of this method was previously analysed theoretically in [23].

The $g_{DG}$ correction function [13] is provided by
(14)$$g_{DG,L}=\frac{{\left(-1\right)}^{k}}{2}\left(L_{k}-L_{k+1}\right),$$
(15)$$g_{DG,R}=\frac{1}{2}\left(L_{k}+L_{k+1}\right),$$
where $L_{k}$ is the Legendre polynomial of order *k*. When *k* is 4, the expressions of the Legendre polynomials are $L_{4}=\left(35x^{4}-30x^{2}+3\right)/8$ and $L_{5}=\left(63x^{5}-70x^{3}+15x\right)/8$.

## 3. CPR (Q>P) Method with Subcell CNNW2 Limiting

A subcell CNNW2 limiting scheme was proposed by Zhu et al. [21]. Cells in the flow field are divided into troubled cells and smooth cells by shock indicators such as TVB [25] or MDHE [26]. Then, the smooth cells are solved by the CPR (Q>P) method and the troubled cells are solved by the CNNW2 scheme. The CPR (Q>P) method with subcell CNNW2 limiting needs to calculate the common numerical flux at the cell interfaces. If the left side of the interface is a troubled cell and the right side is a smooth cell, then the left value in the numerical flux is provided by the nonuniform nonlinear weighted (NNW) scheme. The right value is provided by the Lagrange interpolation of CPR.

This paper focuses on the energy stability of the CNNW2 scheme. The construction of the CNNW2 scheme for the 1D convection equation is introduced in Section 3.1; see [21] for details. In the following, we take three solution points for each cell as an example.

### 3.1. CNNW2 Scheme

The computational domain [a,b] is divided into *N* cells, and the length of each cell is *h*. Each cell is divided into *k* subcells, and the number of subcells is consistent with the number of solution points. Subcells are obtained based on Gauss quadrature weights. Considering the cell stencil of three solution points, the corresponding state variables are $u_{i},i=1,2,3$, as shown in Figure 4. Here, $u_{A}$ and $u_{B}$ are the values at the subcell interfaces and $d_{i},i=1,2,3,4,5,6$ is the distance between the solution points and the flux points. The right value of $u_{A}$ and the left value of $u_{B}$ are obtained by the following method.

(1) The intermediate values $u_{A}^{\left(1\right)}$, $u_{B}^{\left(1\right)}$ are obtained by an inverse distance weighted interpolation:(16)$$\omega_{1}=\frac{\left(1/d_{2}\right)}{\left(1/d_{2}\right)+\left(1/d_{3}\right)},\;\omega_{2}=\frac{\left(1/d_{3}\right)}{\left(1/d_{2}\right)+\left(1/d_{3}\right)},\;u_{A}^{\left(1\right)}=\omega_{1}u_{1}+\omega_{2}u_{2},$$
(17)$$\omega_{3}=\frac{\left(1/d_{4}\right)}{\left(1/d_{4}\right)+\left(1/d_{5}\right)},\;\omega_{4}=\frac{\left(1/d_{5}\right)}{\left(1/d_{4}\right)+\left(1/d_{5}\right)},\;u_{B}^{\left(1\right)}=\omega_{3}u_{2}+\omega_{4}u_{3},$$
where $\omega_{i}$ is the interpolation weight.

(2) The gradient ∂u/∂ξ is calculated by
(18)$$\frac{\partial u}{\partial \xi }=\omega_{5}{\left(\frac{\partial u}{\partial \xi }\right)}^{\left(1\right)}+\omega_{6}{\left(\frac{\partial u}{\partial \xi }\right)}^{\left(2\right)}=\omega_{5}\frac{u_{2}-u_{A}^{\left(1\right)}}{d_{3}}+\omega_{6}\frac{u_{B}^{\left(1\right)}-u_{2}}{d_{4}},$$
where
(19)$$\omega_{5}=\frac{\left(1/d_{3}\right)}{\left(1/d_{3}\right)+\left(1/d_{4}\right)},\;\omega_{6}=\frac{\left(1/d_{4}\right)}{\left(1/d_{3}\right)+\left(1/d_{4}\right)}.$$

(3) With the gradient ∂u/∂ξ and $u_{2}$, $u_{A}$ and $u_{B}$ are recalculated by
(20)$$u_{A}^{\left(2\right)}=u_{2}-\frac{\partial u}{\partial \xi }d_{3},\;u_{B}^{\left(2\right)}=u_{2}+\frac{\partial u}{\partial \xi }d_{4},$$

(4) The gradient is limited in order to control numerical oscillations.
(21)$$u_{A}^{R}=u_{2}-\phi \frac{\partial u}{\partial \xi }d_{3},\;u_{B}^{L}=u_{2}+\phi \frac{\partial u}{\partial \xi }d_{4},\;\phi =\min\left(\lim\left(u_{A}^{\left(2\right)}\right),\lim\left(u_{B}^{\left(2\right)}\right)\right),$$
where the limiting function is defined as
(22)$$\lim\left(u\right)=\left\{\begin{array}{l} \min\left(1,\frac{M-u_{2}}{u-u_{2}}\right),\;\mathrm{if}\;u>u_{2},\\ \min\left(1,\frac{m-u_{2}}{u-u_{2}}\right),\;\mathrm{if}\;u<u_{2},\\ 1,\;\mathrm{if}\;u=u_{2}, \end{array}\right.,$$
where $m=\min\left(u_{1},u_{2},u_{3}\right),\;M=\max\left(u_{1},u_{2},u_{3}\right)$.

Through NNW interpolation, the left and right values of a second-order polynomial with a limiter on the subcell interface can be obtained; thus, the numerical flux at each interface can be obtained. Finally, the second-order difference operator is used to approximate the spatial derivative for the numerical flux.
(23)$$\frac{\partial f}{\partial \xi }=\frac{{\hat{f}}_{n,fp_{j+1}}-{\hat{f}}_{n,fp_{j}}}{\xi_{fp_{j+1}}-\xi_{fp_{j}}},$$
where $\xi_{fp_{j}}$ is the coordinate of the $fp_{j}$ flux point. It should be pointed out that if ϕ=1, the scheme is equivalent to a linear scheme without a limiter. If ϕ=0, the scheme is reduced to a first-order scheme.

### 3.2. Proof of the Linear Energy Stability of the CNNW2 Scheme

Consider the 1D scalar convection equation
(24)$$\frac{\partial u}{\partial t}+\frac{\partial u}{\partial x}=0.$$ The linear energy stability of the scheme is proven by the energy estimation method [24], and periodic boundary conditions are used. The energy norm change rate of the whole physical space and the computational space has the following relationship:(25)$$\frac{\partial }{\partial t}\int_{a}^{b}u^{2}dx=\frac{\partial }{\partial t}\sum_{n=1}^{N}\int_{x_{n}}^{x_{n+1}}u^{2}dx=\frac{h}{2}\frac{\partial }{\partial t}\sum_{n=1}^{N}\int_{-1}^{1}u^{2}d\xi .$$ According to the above equation, in order to calculate the rate of change of the energy norm over time in the whole physical space, we can consider the energy norm change rate of the *n*th computational cell.

By using the CNNW2 scheme, the spatial semi-discrete scheme of the *n*th computational cell is
(26)$$\begin{array}{c} \frac{\partial u_{n,1}}{\partial t}=\frac{{\hat{f}}_{n,fp1}-{\hat{f}}_{n,fp2}}{d_{1}+d_{2}},\\ \frac{\partial u_{n,2}}{\partial t}=\frac{{\hat{f}}_{n,fp2}-{\hat{f}}_{n,fp3}}{d_{3}+d_{4}},\\ \frac{\partial u_{n,3}}{\partial t}=\frac{{\hat{f}}_{n,fp3}-{\hat{f}}_{n,fp4}}{d_{5}+d_{6}}. \end{array}$$

The numerical flux adopts the upwind flux [2], and the numerical flux at each interface can be obtained:(27)$$\begin{array}{cc} \; & {\hat{f}}_{n,fp1}=u_{n-1,3}+\frac{d_{6}^{2}}{d_{5}+d_{6}}\frac{u_{n-1,3}-u_{n-1,2}}{d_{4}+d_{5}}+\frac{d_{5}d_{6}}{d_{5}+d_{6}}\frac{u_{n,1}-u_{n-1,3}}{d_{1}+d_{6}},\\ \; & {\hat{f}}_{n,fp2}=u_{n,1}+\frac{d_{2}^{2}}{d_{1}+d_{2}}\frac{u_{n,1}-u_{n-1,3}}{d_{1}+d_{6}}+\frac{d_{1}d_{2}}{d_{1}+d_{2}}\frac{u_{n,2}-u_{n,1}}{d_{2}+d_{3}},\\ \; & {\hat{f}}_{n,fp3}=u_{n,2}+\frac{d_{4}^{2}}{d_{3}+d_{4}}\frac{u_{n,2}-u_{n,1}}{d_{2}+d_{3}}+\frac{d_{3}d_{4}}{d_{3}+d_{4}}\frac{u_{n,3}-u_{n,2}}{d_{4}+d_{5}},\\ \; & {\hat{f}}_{n,fp4}=u_{n,3}+\frac{d_{6}^{2}}{d_{5}+d_{6}}\frac{u_{n,3}-u_{n,2}}{d_{4}+d_{5}}+\frac{d_{5}d_{6}}{d_{5}+d_{6}}\frac{u_{n+1,1}-u_{n,3}}{d_{1}+d_{6}}. \end{array}$$

Using the Gaussian integral formula, we obtain the energy norm change rate of the *n*th computational cell:(28)$$\begin{array}{cc} \; & \frac{\frac{1}{2}\partial \int_{-1}^{1}u^{2}d\xi }{\partial t}\\ \; & =u_{n,1}\frac{\partial u_{n,1}}{\partial t}w_{1}+u_{n,2}\frac{\partial u_{n,2}}{\partial t}w_{2}+u_{n,3}\frac{\partial u_{n,3}}{\partial t}w_{3}\\ \; & =\left(\frac{d_{5}d_{6}}{d_{5}+d_{6}}\frac{1}{d_{1}+d_{6}}-1-\frac{d_{2}^{2}}{d_{1}+d_{2}}\frac{1}{d_{1}+d_{6}}+\frac{d_{1}d_{2}}{d_{1}+d_{2}}\frac{1}{d_{2}+d_{3}}\right)u_{n,1}^{2}\\ \; & +\left(1+\frac{d_{6}^{2}}{d_{5}+d_{6}}\frac{1}{d_{4}+d_{5}}-\frac{d_{5}d_{6}}{d_{5}+d_{6}}\frac{1}{d_{1}+d_{6}}+\frac{d_{2}^{2}}{d_{1}+d_{2}}\frac{1}{d_{1}+d_{6}}\right)u_{n-1,3}u_{n,1}\\ \; & +\left(-\frac{d_{6}^{2}}{d_{5}+d_{6}}\frac{1}{d_{4}+d_{5}}\right)u_{n-1,2}u_{n,1}+\left(-\frac{d_{1}d_{2}}{d_{1}+d_{2}}\frac{1}{d_{2}+d_{3}}\right)u_{n,2}u_{n,1}\\ \; & +\left(-1-\frac{d_{4}^{2}}{d_{3}+d_{4}}\frac{1}{d_{2}+d_{3}}+\frac{d_{3}d_{4}}{d_{3}+d_{4}}\frac{1}{d_{4}+d_{5}}+\frac{d_{1}d_{2}}{d_{1}+d_{2}}\frac{1}{d_{2}+d_{3}}\right)u_{n,2}^{2}\\ \; & +\left(\frac{d_{4}^{2}}{d_{3}+d_{4}}\frac{1}{d_{2}+d_{3}}+1+\frac{d_{2}^{2}}{d_{1}+d_{2}}\frac{1}{d_{1}+d_{6}}-\frac{d_{1}d_{2}}{d_{1}+d_{2}}\frac{1}{d_{2}+d_{3}}\right)u_{n,1}u_{n,2}\\ \; & +\left(-\frac{d_{3}d_{4}}{d_{3}+d_{4}}\frac{1}{d_{4}+d_{5}}\right)u_{n,3}u_{n,2}+\left(-\frac{d_{2}^{2}}{d_{1}+d_{2}}\frac{1}{d_{1}+d_{6}}\right)u_{n-1,3}u_{n,2}\\ \; & +\left(\frac{d_{3}d_{4}}{d_{3}+d_{4}}\frac{1}{d_{4}+d_{5}}-1-\frac{d_{6}^{2}}{d_{5}+d_{6}}\frac{1}{d_{4}+d_{5}}+\frac{d_{5}d_{6}}{d_{5}+d_{6}}\frac{1}{d_{1}+d_{6}}\right)u_{n,3}^{2}\\ \; & +\left(1+\frac{d_{4}^{2}}{d_{3}+d_{4}}\frac{1}{d_{2}+d_{3}}-\frac{d_{3}d_{4}}{d_{3}+d_{4}}\frac{1}{d_{4}+d_{5}}+\frac{d_{6}^{2}}{d_{5}+d_{6}}\frac{1}{d_{4}+d_{5}}\right)u_{n,2}u_{n,3}\\ \; & +\left(-\frac{d_{4}^{2}}{d_{3}+d_{4}}\frac{1}{d_{2}+d_{3}}\right)u_{n,1}u_{n,3}+\left(-\frac{d_{5}d_{6}}{d_{5}+d_{6}}\frac{1}{d_{1}+d_{6}}\right)u_{n+1,1}u_{n,3}, \end{array}$$
where $u_{n,i},i=1,2,3$ denotes the state variables at the *i*th solution point on the *n*th cell. In addition, $w_{1},w_{2},w_{3}$ are the Gauss quadrature weights corresponding to the solution points.

The energy norm change rate of the whole computational space is
(29)$$\begin{array}{cc} \; & \frac{\frac{1}{2}\sum_{n=1}^{N}\partial \int_{-1}^{1}u^{2}d\xi }{\partial t}\\ \; & =\left(\frac{d_{5}d_{6}}{d_{5}+d_{6}}\frac{1}{d_{1}+d_{6}}-1-\frac{d_{2}^{2}}{d_{1}+d_{2}}\frac{1}{d_{1}+d_{6}}+\frac{d_{1}d_{2}}{d_{1}+d_{2}}\frac{1}{d_{2}+d_{3}}\right)\sum_{n=1}^{N}u_{n,1}^{2}\\ \; & +\left(-1-\frac{d_{4}^{2}}{d_{3}+d_{4}}\frac{1}{d_{2}+d_{3}}+\frac{d_{3}d_{4}}{d_{3}+d_{4}}\frac{1}{d_{4}+d_{5}}+\frac{d_{1}d_{2}}{d_{1}+d_{2}}\frac{1}{d_{2}+d_{3}}\right)\sum_{n=1}^{N}u_{n,2}^{2}\\ \; & +\left(\frac{d_{3}d_{4}}{d_{3}+d_{4}}\frac{1}{d_{4}+d_{5}}-1-\frac{d_{6}^{2}}{d_{5}+d_{6}}\frac{1}{d_{4}+d_{5}}+\frac{d_{5}d_{6}}{d_{5}+d_{6}}\frac{1}{d_{1}+d_{6}}\right)\sum_{n=1}^{N}u_{n,3}^{2}\\ \; & +\left(1+\frac{d_{6}^{2}}{d_{5}+d_{6}}\frac{1}{d_{4}+d_{5}}-2\frac{d_{5}d_{6}}{d_{5}+d_{6}}\frac{1}{d_{1}+d_{6}}+\frac{d_{2}^{2}}{d_{1}+d_{2}}\frac{1}{d_{1}+d_{6}}\right)\sum_{n=1}^{N}u_{n-1,3}u_{n,1}\\ \; & +\left(\frac{d_{4}^{2}}{d_{3}+d_{4}}\frac{1}{d_{2}+d_{3}}+1+\frac{d_{2}^{2}}{d_{1}+d_{2}}\frac{1}{d_{1}+d_{6}}-2\frac{d_{1}d_{2}}{d_{1}+d_{2}}\frac{1}{d_{2}+d_{3}}\right)\sum_{n=1}^{N}u_{n,1}u_{n,2}\\ \; & +\left(1+\frac{d_{4}^{2}}{d_{3}+d_{4}}\frac{1}{d_{2}+d_{3}}-2\frac{d_{3}d_{4}}{d_{3}+d_{4}}\frac{1}{d_{4}+d_{5}}+\frac{d_{6}^{2}}{d_{5}+d_{6}}\frac{1}{d_{4}+d_{5}}\right)\sum_{n=1}^{N}u_{n,2}u_{n,3}\\ \; & +\left(-\frac{d_{2}^{2}}{d_{1}+d_{2}}\frac{1}{d_{1}+d_{6}}\right)\sum_{n=1}^{N}u_{n-1,3}u_{n,2}+\left(-\frac{d_{6}^{2}}{d_{5}+d_{6}}\frac{1}{d_{4}+d_{5}}\right)\sum_{n=1}^{N}u_{n-1,2}u_{n,1}\\ \; & +\left(-\frac{d_{4}^{2}}{d_{3}+d_{4}}\frac{1}{d_{2}+d_{3}}\right)\sum_{n=1}^{N}u_{n,1}u_{n,3}. \end{array}$$

Taking $\sum_{n=1}^{N}\int_{-1}^{1}u^{2}d\xi =E$, the above equation can be converted into
(30)$$\begin{array}{cc} \frac{1}{2}\frac{\partial E}{\partial t} & =\left(a_{1}+b_{1}\right)\sum_{n=1}^{N}\left(u_{n,1}^{2}+u_{n,2}^{2}\right)+\left(-2a_{1}-2b_{1}\right)\sum_{n=1}^{N}\left(u_{n,1}u_{n,2}\right)\\ \; & +\left(c_{1}\right)\sum_{n=1}^{N}\left(u_{n,1}^{2}+u_{n,2}^{2}\right)+\left(-2c_{1}\right)\sum_{n=1}^{N}\left(u_{n,1}u_{n-1,2}\right)\\ \; & +\left(a_{2}+b_{2}\right)\sum_{n=1}^{N}\left(u_{n,1}^{2}+u_{n,3}^{2}\right)+\left(-2a_{2}-2b_{2}\right)\sum_{n=1}^{N}\left(u_{n,1}u_{n-1,3}\right)\\ \; & +\left(c_{2}\right)\sum_{n=1}^{N}\left(u_{n,1}^{2}+u_{n,3}^{2}\right)+\left(-2c_{2}\right)\sum_{n=1}^{N}\left(u_{n,1}u_{n,3}\right)\\ \; & +\left(a_{3}+b_{3}\right)\sum_{n=1}^{N}\left(u_{n,2}^{2}+u_{n,3}^{2}\right)+\left(-2a_{3}-2b_{3}\right)\sum_{n=1}^{N}\left(u_{n,2}u_{n,3}\right)\\ \; & +\left(c_{3}\right)\sum_{n=1}^{N}\left(u_{n,2}^{2}+u_{n,3}^{2}\right)+\left(-2c_{3}\right)\sum_{n=1}^{N}\left(u_{n,2}u_{n-1,3}\right), \end{array}$$
where
(31)$$\begin{array}{cc} \; & a_{1}=-\frac{1}{2}-\frac{1}{2}\frac{d_{4}^{2}}{d_{3}+d_{4}}\frac{1}{d_{2}+d_{3}}-\frac{1}{2}\frac{d_{2}^{2}}{d_{1}+d_{2}}\frac{1}{d_{1}+d_{6}},\\ \; & b_{1}=\frac{d_{1}d_{2}}{d_{1}+d_{2}}\frac{1}{d_{2}+d_{3}},\\ \; & c_{1}=\frac{1}{2}\frac{d_{6}^{2}}{d_{5}+d_{6}}\frac{1}{d_{4}+d_{5}},\\ \; & a_{2}=-\frac{1}{2}-\frac{1}{2}\frac{d_{6}^{2}}{d_{5}+d_{6}}\frac{1}{d_{4}+d_{5}}-\frac{1}{2}\frac{d_{2}^{2}}{d_{1}+d_{2}}\frac{1}{d_{1}+d_{6}},\\ \; & b_{2}=\frac{d_{5}d_{6}}{d_{5}+d_{6}}\frac{1}{d_{1}+d_{6}},\\ \; & c_{2}=\frac{1}{2}\frac{d_{4}^{2}}{d_{3}+d_{4}}\frac{1}{d_{2}+d_{3}},\\ \; & a_{3}=-\frac{1}{2}-\frac{1}{2}\frac{d_{4}^{2}}{d_{3}+d_{4}}\frac{1}{d_{2}+d_{3}}-\frac{1}{2}\frac{d_{6}^{2}}{d_{5}+d_{6}}\frac{1}{d_{4}+d_{5}},\\ \; & b_{3}=\frac{d_{3}d_{4}}{d_{3}+d_{4}}\frac{1}{d_{4}+d_{5}},\\ \; & c_{3}=\frac{1}{2}\frac{d_{2}^{2}}{d_{1}+d_{2}}\frac{1}{d_{1}+d_{6}}, \end{array}$$

Since $d_{1}=d_{6}\;\approx \;0.2254$, $d_{2}=d_{5}\;\approx \;0.3302$ and $d_{3}=d_{4}\;\approx \;0.4444$, $\left|a_{i}\right|\geq \left|b_{i}+c_{i}\right|$ and $a_{i}\leq 0,b_{i}\geq 0,c_{i}\geq 0,i=1,2,3$ are satisfied. According to the average value inequality $a^{2}+b^{2}\geq 2ab$,
(32)$$\begin{array}{cc} \frac{1}{2}\frac{\partial E}{\partial t} & =\left(a_{1}+b_{1}\right)\sum_{n=1}^{N}\left(u_{n,1}^{2}+u_{n,2}^{2}\right)+\left(-2a_{1}-2b_{1}\right)\sum_{n=1}^{N}\left(u_{n,1}u_{n,2}\right)\\ \; & +\left(c_{1}\right)\sum_{n=1}^{N}\left(u_{n,1}^{2}+u_{n,2}^{2}\right)+\left(-2c_{1}\right)\sum_{n=1}^{N}\left(u_{n,1}u_{n-1,2}\right)\\ \; & +\left(a_{2}+b_{2}\right)\sum_{n=1}^{N}\left(u_{n,1}^{2}+u_{n,3}^{2}\right)+\left(-2a_{2}-2b_{2}\right)\sum_{n=1}^{N}\left(u_{n,1}u_{n-1,3}\right)\\ \; & +\left(c_{2}\right)\sum_{n=1}^{N}\left(u_{n,1}^{2}+u_{n,3}^{2}\right)+\left(-2c_{2}\right)\sum_{n=1}^{N}\left(u_{n,1}u_{n,3}\right)\\ \; & +\left(a_{3}+b_{3}\right)\sum_{n=1}^{N}\left(u_{n,2}^{2}+u_{n,3}^{2}\right)+\left(-2a_{3}-2b_{3}\right)\sum_{n=1}^{N}\left(u_{n,2}u_{n,3}\right)\\ \; & +\left(c_{3}\right)\sum_{n=1}^{N}\left(u_{n,2}^{2}+u_{n,3}^{2}\right)+\left(-2c_{3}\right)\sum_{n=1}^{N}\left(u_{n,2}u_{n-1,3}\right)\\ \; & \leq 2\left(a_{1}+b_{1}\right)\sum_{n=1}^{N}\left(u_{n,1}u_{n,2}\right)+\left(2c_{1}\right)\sum_{n=1}^{N}\left(u_{n,1}u_{n-1,2}\right)\\ \; & +\left(-2a_{1}-2b_{1}\right)\sum_{n=1}^{N}\left(u_{n,1}u_{n,2}\right)+\left(-2c_{1}\right)\sum_{n=1}^{N}\left(u_{n,1}u_{n-1,2}\right)\\ \; & +2\left(a_{2}+b_{2}\right)\sum_{n=1}^{N}\left(u_{n,1}u_{n-1,3}\right)+\left(2c_{2}\right)\sum_{n=1}^{N}\left(u_{n,1}u_{n,3}\right)\\ \; & +\left(-2a_{2}-2b_{2}\right)\sum_{n=1}^{N}\left(u_{n,1}u_{n-1,3}\right)+\left(-2c_{2}\right)\sum_{n=1}^{N}\left(u_{n,1}u_{n,3}\right)\\ \; & +2\left(a_{3}+b_{3}\right)\sum_{n=1}^{N}\left(u_{n,2}u_{n,3}\right)+\left(2c_{3}\right)\sum_{n=1}^{N}\left(u_{n,2}u_{n-1,3}\right)\\ \; & +\left(-2a_{3}-2b_{3}\right)\sum_{n=1}^{N}\left(u_{n,2}u_{n,3}\right)+\left(-2c_{3}\right)\sum_{n=1}^{N}\left(u_{n,2}u_{n-1,3}\right)\\ \; & =0, \end{array}$$

Substituting the above equation into Equation (Equation 25), the energy norm change rate of the whole physical space is obtained as
(33)$$\begin{array}{cc} \; & \frac{\partial }{\partial t}\int_{a}^{b}u^{2}dx=\frac{h}{2}\frac{\sum_{n=1}^{N}\partial \int_{-1}^{1}u^{2}d\xi }{\partial t}\leq 0. \end{array}$$ Since *h* is positive, the above result shows that the CNNW2 scheme is linear energy stable with ϕ=1. In Appendix A, we prove the energy stability property of the first-order CNNW2 scheme.

It is worth mentioning that, the linear energy stability of the sucell linear CNNW2 scheme for fifth-order CPR can be proved in a similar deductive procedure.

## 4. Numerical Experiments

To simplify the description, the linear CNNW2 scheme is named CNNW2−L, the nonlinear CNNW2 scheme is named CNNW2−N. Since the CPR based on subcell limiting is in fact a hybrid CPR and CNNW scheme, we take the same abbreviation HCCS as in [21]. Then, the CPR (Q>P) method with subcell linear CNNW2 limiting is named HCCS−L2, and the CPR (Q>P) method with subcell nonlinear CNNW2 limiting is named HCCS−N2.

The linear energy stability of the CPR (Q>P), CNNW2−L and HCCS−L2 schemes are studied in the context of 1D [13] and 2D linear convection problems [27]. An isentropic vortex case is used to analyse the errors of the CPR (Q > P), HCCS−L2 and HCCS−N2 schemes. The nonlinear stability of the CPR (Q=P), CPR (Q>P) and HCCS−N2 schemes are studied in the context of 2D subsonic flow over a cylinder [28]. The nonlinear stability of the HCCS−N2 scheme is studied in the context of the 2D Kelvin–Helmholtz (KH) instability problem [29] and transonic flows around the NACA0012 airfoil [30]. The $g_{DG}$ correction function and the explicit third-order TVD Runge–Kutta method [31] are used in this section. The $L^{2}$ energy is expressed as
(34)$${\left(\int_{\Omega }u^{2}dx\right)}^{\frac{1}{2}}$$
where Ω is the computational domain of physical space.

If the calculation does not blow up, then the numerical method is stable. Otherwise, if the density or pressure becomes negative, resulting in calculation blowing up, the numerical method is unstable.

### 4.1. Linear Energy Stability Test

#### 4.1.1. 1D Linear Convection Equation

The 1D linear convection Equation (Equation 24) is used. The computational domain is defined in [−1, 1]. The polynomial order is 3, and the number of grid cells is 100. Using upwind flux and periodic boundary conditions, the compution time t is 20 and time steps are determined by CFL = 0.1. This case uses the TVB indicator with adjustable parameter M=1. The initial condition is set as follows:(35)$$u\left(x,0\right)=e^{-20x^{2}}.$$

Figure 5a shows the numerical solutions of the CPR (Q>P), CNNW2−L and HCCS−L2 schemes. Figure 5b shows the $L^{2}$ energy of these three schemes over time. The $L^{2}$ energy of the CPR (Q>P) method remains essentially constant over time. This is because the CPR (Q>P) method has high spatial accuracy and low numerical dissipation. The results of the other two schemes decrease gradually over a bounded range, which is because the CNNW2 scheme has more numerical dissipation than the CPR (Q>P) method. Thus, the three schemes are energy-stable. Figure 6 shows the evolution of troubled cells in the HCCS−L2 scheme, where red represents troubled cells.

#### 4.1.2. 2D Linear Convection Equation

Consider the 2D linear convection equation of
(36)$$\frac{\partial u}{\partial t}+\frac{\partial u}{\partial x}+\frac{\partial u}{\partial y}=0.$$ The computational domain is defined in [−5,5]×[−5,5]. The polynomial order is 3. 60×60 uniform quadrilateral grids are used for the computations. Using upwind flux and periodic boundary conditions, the compution time t is 20 with time steps calculated by CFL = 0.1. This case uses the TVB indicator with adjustable parameter M=0.6. The initial condition is set as follows:(37)$$u\left(x,y,0\right)=e^{-\left(x^{2}+y^{2}\right)}.$$

Figure 7, Figure 8 and Figure 9 respectively show the numerical solutions of the CPR (Q>P), CNNW2−L and HCCS−L2 schemes along with the changes in the $L^{2}$ energy norm over time. The results are consistent with the analysis based on the 1D convection equation. The three schemes are energy-stable, and the CPR (Q>P) method is the highest in accuracy. Figure 9c shows the distribution of troubled cells for the HCCS−L2 scheme, where troubled cells are marked with the red color.

### 4.2. Nonlinear Stability Test

#### 4.2.1. Isentropic Vortex Test

To analyse the errors of the CPR (Q>P), HCCS−L2 and HCCS−N2 schemes, we solve the isentropic vortex problem. In this case, an isentropic vortex disturbance is added to an uniform flow. The uniform flow is set to $\left(\rho_{\infty },u_{\infty },v_{\infty },p_{\infty }\right)=\left(1.0,1.0,0.0,1.0\right)$, and $T_{\infty }=p_{\infty }/\rho_{\infty }$. The initial conditions of the vortex are set as follows:(38)$$\begin{array}{cc} \; & \Delta u=-\left(y-y_{c}\right)\frac{\varepsilon }{2\pi }\exp\left(\frac{1-r^{2}}{2}\right),\\ \; & \Delta v=\left(x-x_{c}\right)\frac{\varepsilon }{2\pi }\exp\left(\frac{1-r^{2}}{2}\right),\\ \; & \Delta T=-\frac{(\gamma -1)\varepsilon }{8\gamma \pi^{2}}\exp\left(1-r^{2}\right), \end{array}$$
where $r=\sqrt{{\left(x-x_{c}\right)}^{2}+{\left(y-y_{c}\right)}^{2}}$, the vortex centre $\left(x_{c},y_{c}\right)=\left(0.0,0.0\right)$, and the vortex strength ϵ=5.0. The initial conditions of the flow field are as follows:(39)$$\left(\rho ,u,v,p\right)=[{\left(T_{\infty }+\Delta T\right)}^{\frac{\gamma }{\gamma -1}},u_{\infty }+\Delta u,v_{\infty }+\Delta v,{\left(T_{\infty }+\Delta T\right)}^{\frac{1}{\gamma -1}}]$$ The computational domain is [−10,10]×[−10,10]. The LLF flux and periodic boundary conditions are used. The compution time t is 0.1 and time steps are calculated using CFL = 0.1.

Table 1 shows the errors of the CPR (Q>P), HCCS−L2 and HCCS−N2 schemes. The errors of the CPR (Q>P) method are the smallest, while the errors of the HCCS−N2 scheme are the largest.

#### 4.2.2. Subsonic Flow over a Cylinder

The cylinder radius is 0.5, and the far field is ten times the cylinder diameter. The simulation is run at a freestream Mach number of 0.2 with 5th-order accuracy. 178 cells are distributed in the circumferential direction, while 54 cells are distributed in the radial direction. Using the LLF flux, the compution is conducted until t = 30 with time steps determined by CFL = 0.3. The inviscid wall boundary condition is imposed on the wall surface, and the far-field boundary condition is imposed on the far field. This case uses the MDHE indicator with an adjustable parameter a=0.0005. When solving this problem with straight-sided quadrilateral grids, a pair of wake vortices appears at the rear end of the cylinder.

Figure 10 shows the density contours calculated by the CPR (Q=P) and CPR (Q>P) schemes. It can be seen that the CPR (Q=P) scheme blows up at t=14.25 due to the effect of aliasing errors. Figure 10a shows the flow field before the blow up. The CPR (Q>P) scheme simulates stably to the end.

Figure 11 shows the density contour and the distribution of troubled cells calculated by the HCCS−N2 scheme with troubled cells represented by red color. The CPR (Q>P) scheme with subcell nonlinear CNNW2 limiting is shown to be still nonlinearly stable.

#### 4.2.3. 2D Kelvin-Helmholtz Instability Problem

The initial conditions are set as follows:(40)$$\begin{array}{cc} \rho (x,y) & =\frac{1}{2}+\frac{3}{4}B,\;p\left(x,y\right)=1,\\ u(x,y) & =\frac{1}{2}\left(B-1\right),\;v\left(x,y\right)=\frac{1}{10}\sin\left(2\pi x\right), \end{array}$$
where B=tanh(15y+7.5)−tanh(15y−7.5). The computational domain is defined in [−1,1]×[−1,1]. The polynomial order is 7, and the number of grid cells is 64×64. Using the LLF flux and periodic boundary conditions, the compution time t is 10 with time steps calculated by CFL = 0.3. The Reynolds number goes to infinity, and the Mach number is approximately 0.6. This case uses the MDHE indicator with an adjustable parameter a=0.5.

Figure 12a shows the density contour calculated by the HCCS−N2 scheme. The high resolution of this scheme enables it to capture small-scale features. Figure 12b shows the distribution of troubled cells (in red).

#### 4.2.4. 2D Transonic Flow around the NACA0012 Airfoil

The initial condition of transonic flow around the NACA0012 airfoil is a freestream flow condition with Mach number $Ma_{\infty }=0.8$ and angle of attack $\alpha =1.25^{\circ }$. The inviscid wall boundary condition is imposed on the wall surface, and the far-field boundary condition is imposed on the far field. The numbers of grid cells distributed in the circumferential and radial directions are 120 and 80, respectively, which are generated by solving a partial differential equation, as shown in Figure 13. The polynomial order is 2, and the compution time t is 50 with time steps determined by CFL = 0.5. This case uses the MDHE indicator with an adjustable parameter a=0.0008.

Figure 14 shows the density contour, pressure contour and Mach number contour calculated by the HCCS−N2 scheme. This scheme can better capture the relatively strong shock waves on the upper surface of the airfoil and relatively weak shock waves on the lower surface.

Figure 15 shows the pressure coefficient distribution on the airfoil surface calculated by the HCCS−N2 scheme. While there are slight differences in the shock wave regions, the result is in good agreement with the reference solution in the other regions.

## 5. Conclusions

This paper addresses the energy stability of the CPR method with staggered flux points (CPR (Q>P)) based on second-order subcell limiting. The linear CNNW2 scheme with ϕ = 1 is proven to be energy stable. Through numerical tests of 1D and 2D linear convection equations, the results show that linear CNNW2 and CPR (Q>P) method with subcell linear CNNW2 limiting are energy-stable. Through numerical tests of 2D subsonic flow over a cylinder show that the CPR (Q>P) method has better nonlinear stability than the CPR (Q=P) method. The results of 2D Kelvin–Helmholtz (KH) instability problem and transonic flows around the NACA0012 airfoil indicate that the CPR (Q>P) with subcell CNNW2 limiting has good properties in both shock capturing and nonlinear stability. It also shows that the CPR (Q>P) method with subcell nonlinear CNNW2 limiting is nonlinearly stable.

## Figures and Tables

**Figure 1 entropy-25-00729-f001:**
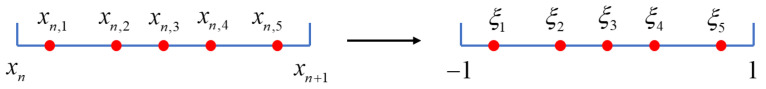
A linear transformation between a physical cell and its computational cell.

**Figure 2 entropy-25-00729-f002:**
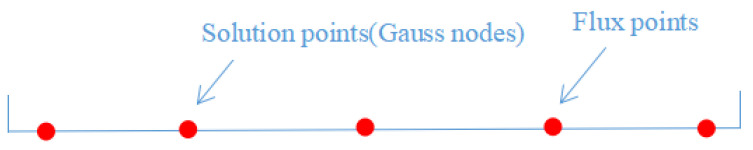
Location of solution points and flux points of the original CPR method (p4).

**Figure 3 entropy-25-00729-f003:**
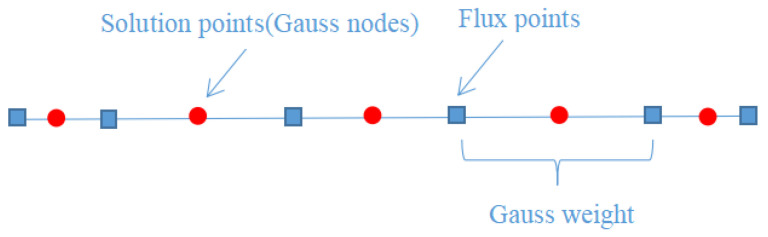
Location of solution points and flux points of the CPR (Q>P) method (p4).

**Figure 4 entropy-25-00729-f004:**
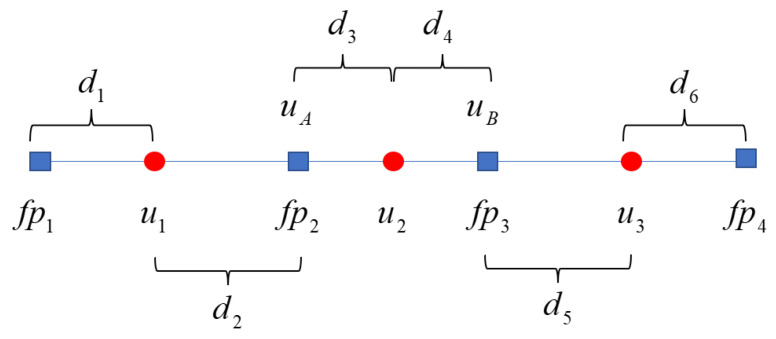
The position relationship between the solution points (red circles) and the flux points (blue squares) in the CNNW2 scheme. Here, $fp_{j}$ denotes the *j*th flux point, *j* = 1, 2, 3, 4.

**Figure 5 entropy-25-00729-f005:**
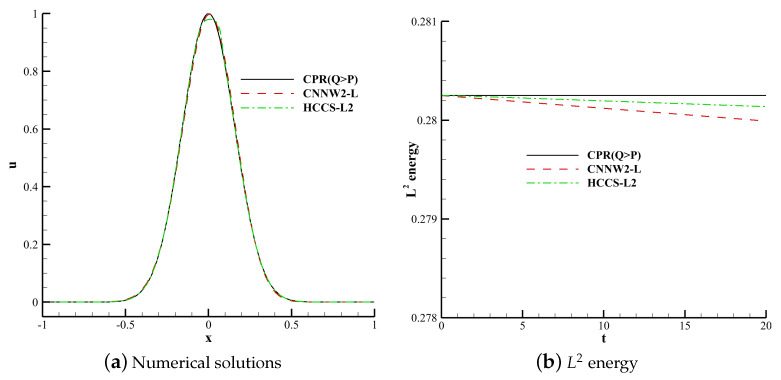
Numerical solutions and $L^{2}$ energy of the CPR (Q>P), CNNW2−L and HCCS−L2 schemes.

**Figure 6 entropy-25-00729-f006:**
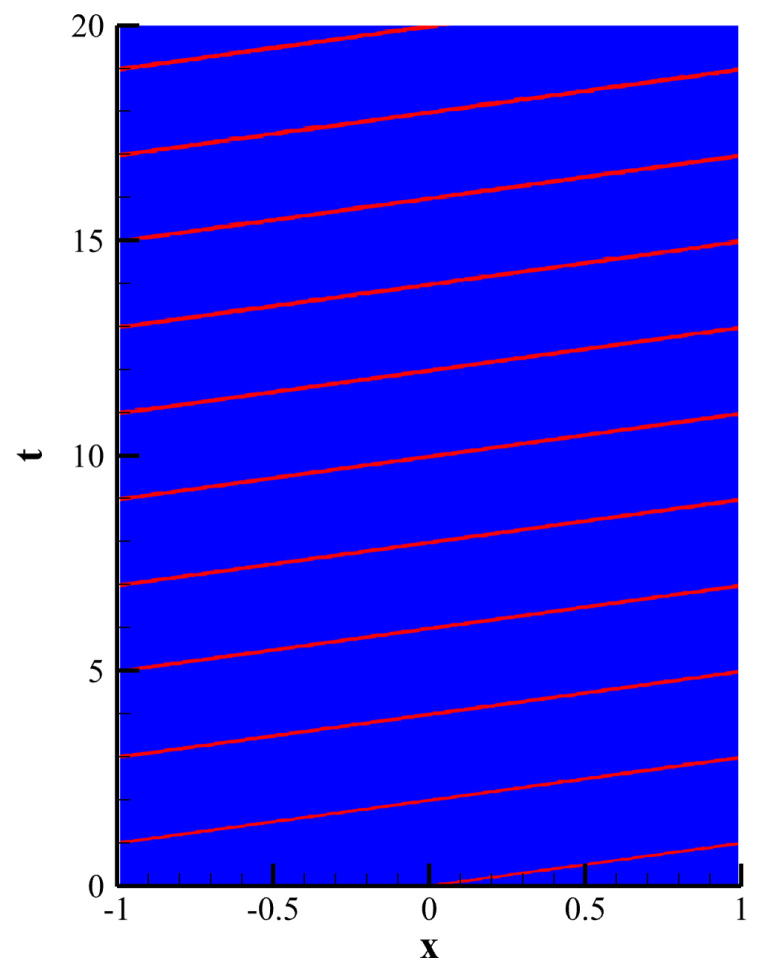
Distributions of troubled cells corresponding to the HCCS−L2 scheme.

**Figure 7 entropy-25-00729-f007:**
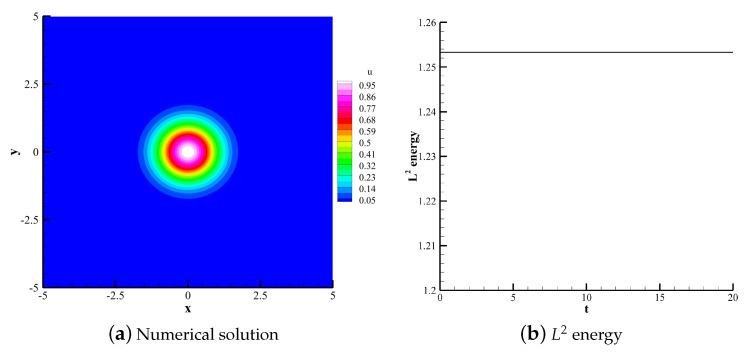
Numerical solution and $L^{2}$ energy of the CPR (Q>P) scheme.

**Figure 8 entropy-25-00729-f008:**
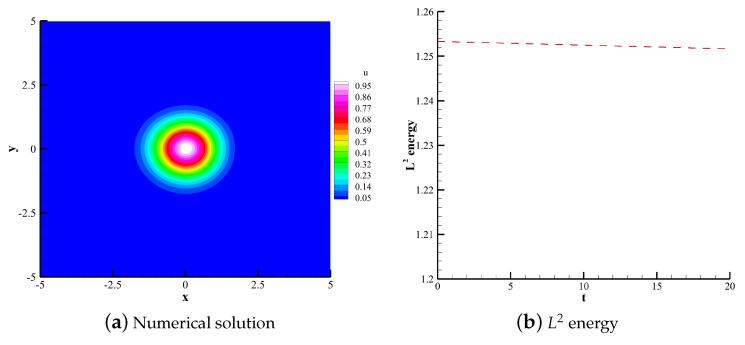
Numerical solution and $L^{2}$ energy of the CNNW2−L scheme.

**Figure 9 entropy-25-00729-f009:**
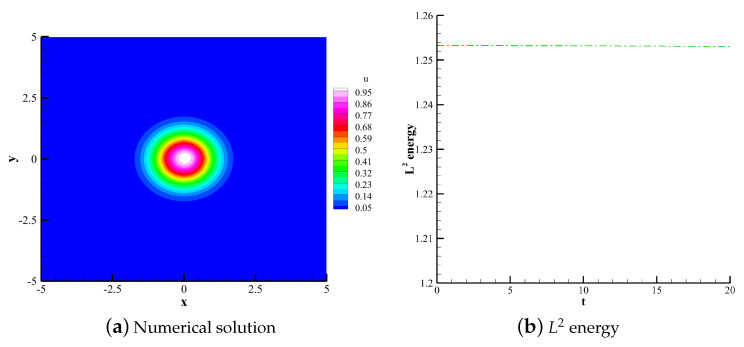
Numerical solution, $L^{2}$ energy and troubled cell distribution of the HCCS−L2 scheme.

**Figure 10 entropy-25-00729-f010:**
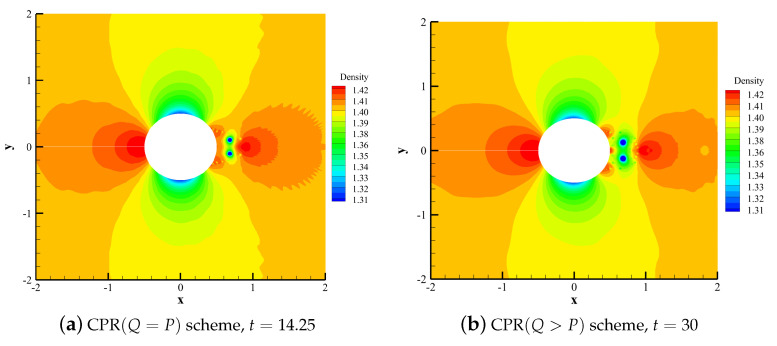
Density contours of the CPR (Q=P) and CPR (Q>P) schemes.

**Figure 11 entropy-25-00729-f011:**
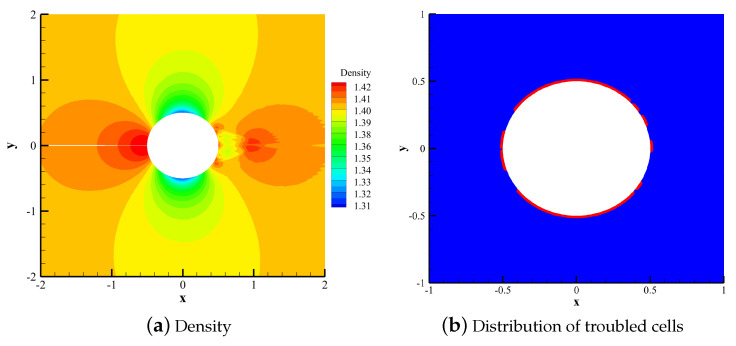
Density contour and troubled cell distributions in the HCCS−N2 scheme.

**Figure 12 entropy-25-00729-f012:**
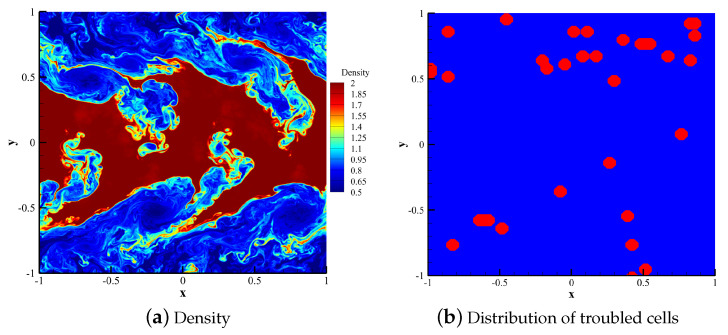
Density contour and troubled cell distributions in the HCCS−N2 scheme.

**Figure 13 entropy-25-00729-f013:**
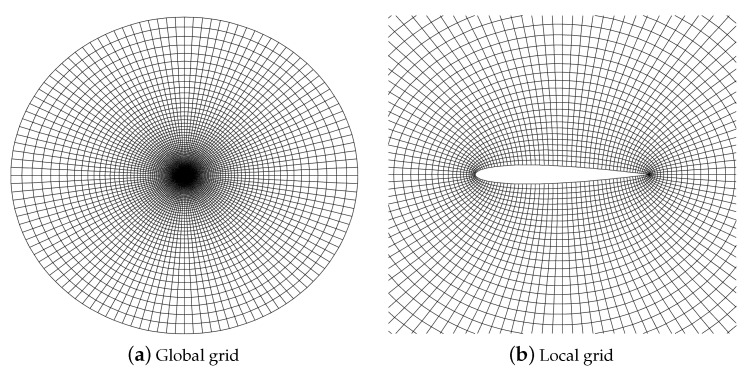
NACA0012 airfoil grid.

**Figure 14 entropy-25-00729-f014:**
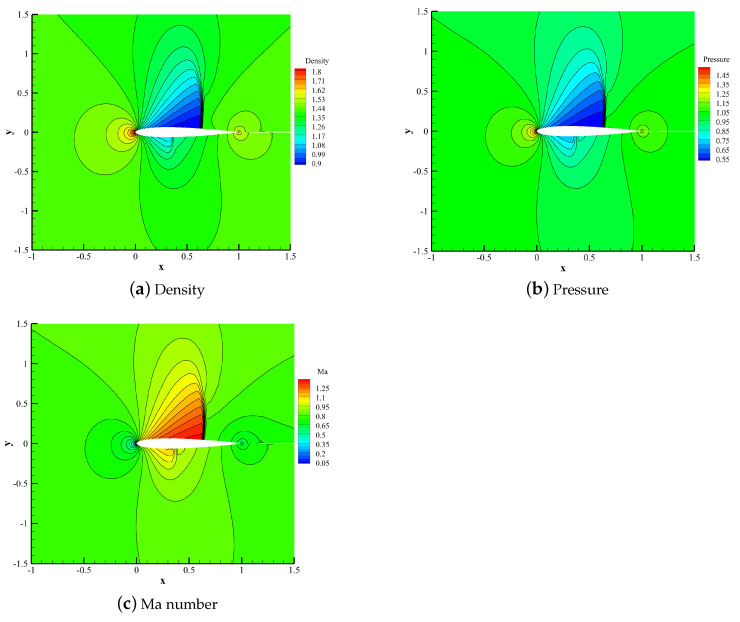
Density contour, pressure contour, and Mach number contour of the NACA0012 airfoil.

**Figure 15 entropy-25-00729-f015:**
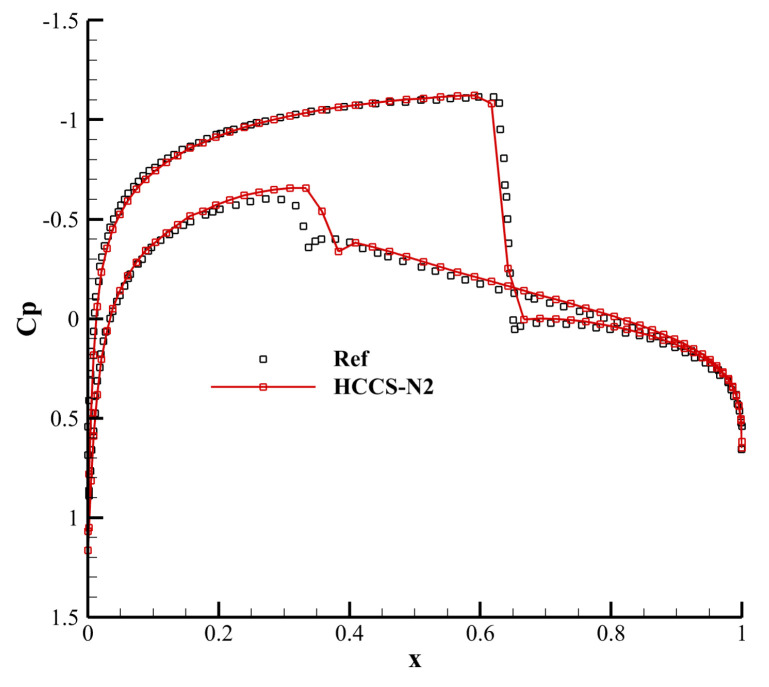
Pressure coefficient distribution on the airfoil surface.

**Table 1 entropy-25-00729-t001:** Errors of three schemes.

Error	Mesh	CPR (Q > P)	HCCS−L2	HCCS−N2
$L_{1}$ error	20 × 20	1.01 × 10${}^{-5}$	4.87 × 10${}^{-5}$	6.73 × 10${}^{-5}$
	40 × 40	7.07 × 10${}^{-7}$	1.21 × 10${}^{-5}$	1.54 × 10${}^{-5}$
	80 × 80	4.05 × 10${}^{-8}$	1.80 × 10${}^{-5}$	2.50 × 10${}^{-6}$
$L_{\infty }$ error	20 × 20	6.52 × 10${}^{-4}$	5.71 × 10${}^{-3}$	9.07 × 10${}^{-3}$
	40 × 40	7.55 × 10${}^{-5}$	2.10 × 10${}^{-3}$	3.67 × 10${}^{-3}$
	80 × 80	4.21 × 10${}^{-6}$	6.32 × 10${}^{-4}$	1.44 × 10${}^{-3}$

## Data Availability

Not applicable.

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
