# Peer review of "Energy Stability Property of the CPR Method Based on Subcell Second-Order CNNW Limiting in Solving Conservation Laws"

_entropy, 2023, doi:10.3390/e25050729_

Round 1
Reviewer 1 Report
Review of paper "Energy stability property of the CPR method based on subcell second-order CNNW limiting in solving conservation laws" by Ran Liu, Zhenguo Yan, Huajun Zhu, Feiran Jia, Xinlong Feng.
The paper focuses on the stability property of the correction procedure via the reconstruction (CPR) method. The staggered flux points based on second-order subcell limiting are used and a shock indicator is used to detect troubled cells containing possible discontinuities. The troubled cells are calculated by the second-order subcell compact nonuniform nonlinear weighted scheme. The linear energy stability of the linear scheme is proved theoretically for the 1D problem. The numerical method is also applied for various numerical experiments both in 1D as well as 2D, where the CNNW2 scheme and CPR method is shown to be energy stable.
The paper is structure into 5 sections, after nicely written Introduction the authors in section 2 focus on description of formulation of 2D Euler equations(2.1) and CPR method is
presented for 1D problem in 2.2. Section 3 focuses on limiting. Section 4 presents numerical results and the last section presents conclusions.
I found the paper to be interesting, but it requires several modifications before publishing, see the list below. Most importantly the proof in section 3.2 is incorrect. Overall, I recommend a major revision of the paper.
List of revisions needed:
1) Revise abstract - on 10 lines of the abstract it uses “staggered flux points” 6 times, which does not bring better explanation, but it is rather confusing.
2) The first sentence of the Introduction is in my opinion incorrect “In recent years ... corresponding CFD commercial software has also been launched”. The history of commercial codes can not be specified as “recent years”.
3) Page 2, Line 64 sentence “CNNW schemes include …. and CNNW2” is really confusing and it should be left out (or reformulated).
4) Page 3, line 88 - “U is the vector of conservative variables”
5) Section 2.1 should be equipped with at least one citation.
6) Section 2.2 requires major revisions: meaning of several symbols (as u, f, \xi) is unexplained in the text, similarly as meaning of standard cell. Symbols u and f should be probably vectors. Line 94 “Upon transformation to standard cells” should be better explained. Line 98 mentions “the solution point” - but the solution point was neither mentioned before nor explained till the moment.
Notation of CPR(Q>P) and CPR(Q=P) is confusing and its explanation should be improved - lines 107-11.
Eq. (13) and (14) presents function gDG with L/R without any explanation or citation, and without specifying their argument - however, the Legendre polynomials are in general defined over interval [-1,1], so explanation at what point these are evaluated should be given.
7) The theoretical proof in section 3.2 seems incorrect:
-
Eqs. (25-29) are presented on 4 pages, but such a length makes the proof impossible to check! E.g. Eq. 26 is just copy and paste of Eq. 25 - with only summation added. The authors also did not write almost any textual explanation of the individual steps. In particular, eq (25) should be explained.
-
In particular in eq (25) it is unclear how the partial derivative with respect to time was treated? Probably a discrete variant of eq. (23) was involved, but a proper explanation is missing.
-
It is acceptable to use integration in terms of \xi in (25), but the use of the same within (26), where summation over all cell is used leading to different definition of \xi over each cell should be used
-
Line 177-178: “According to the triangle inequality … “. I do not think the proof is correct. For instance the 1st and 4th line of (30) is always >= 0 as c2 >= and (u1 - u3)^2 >=0. Moreover, the use of “triangle inequality” in my opinion does not explain all the inequalities used. It seems that there were several inequalities used, but - in fact several triangle inequalities were used, but the formula (30) - more precisely the term X on the rhs. of <= X = 0 is not necessarily to be provided as the left hand side of the inequality is in fact in the form (a - b)^2
-
Several sentences are confusing:
line 170 - 171: “The Riemann flux adopts the upwind flux” ?
line 176 - 177: “Due to d1 = … “ . It seems that there is something missing in the sentence.
8) Sentence 182-184 is confusing, all the used abbreviations on lines 182-190 should be reconsidered.
9) Caption of 5 should be improved - troubled cells are marked in blue or red?
Reviewer 2 Report
Review on manuscript entitled “Energy stability property of the CPR method based on subcell second-order CNNW limiting in solving conservation laws”.
This paper considers the energy stability property of the correction procedure via the reconstruction (CPR) method with staggered flux points based on second-order subcell limiting. Some cells are calculated by the second-order subcell compact nonuniform nonlinear weighted (CNNW2), while others are calculated by CPR method with staggered flux points. A few test cases are shown. Work is consistent and paper organization is logical.
However, I have some recommendations for improvements, mainly directed to numerical experiments, which are not well presented. These are:
1) Solution stability is predefined in the beginning by the conservation low formulation.
2) Please define the energy criterion that you are using as an equation.
3) Numerical experiments in chapter 4 must be better explained. Implementation is unclear. In my opinion each test case must be explained in details, especially numerical implementation. Please, enlarge the description for each test case.
4) Error analysis of the results must be made. With and without CNNW2.
In conclusion, this method improvement is interesting, but verification process description must be improved.
Round 2
Reviewer 1 Report
As author correctly addressed all the needed corrections, I recommend the paper to be accepted.
Reviewer 2 Report
I accept the improvements made so far.
Still believe that Conclusion must be enlarged.